# Characteristics and Species Diversity of Semi-Natural Plant Communities on Langqi Island

**DOI:** 10.3390/biology13010011

**Published:** 2023-12-24

**Authors:** Yang Liu, Yunteng Huang, Yingxue Wang, Chunxiao Wang, Zixin Xiao, Siyuan Shen, Jiyi Zeng, Chuanyuan Deng

**Affiliations:** College of Landscape Architecture and Art, Fujian Agriculture and Forestry University, Fuzhou 350100, China; liuyoung0429@126.com (Y.L.); 13970739658@163.com (Y.H.); wangyingxue1019@163.com (Y.W.); 13125369987@163.com (C.W.); x17346771673@163.com (Z.X.); ssy13721075909@163.com (S.S.); l1032352370@163.com (J.Z.)

**Keywords:** Langqi Island, semi-natural plant communities, community characteristics, species diversity, redundancy analysis, mixed planting

## Abstract

**Simple Summary:**

This study investigated the characteristics and species diversity of semi-natural plant communities on Langqi Island, a coastal island in Fujian Province, China. Additionally, we explored the environmental factors that influenced species diversity and community distribution. The results revealed that the semi-natural plant communities on Langqi Island had rich plant species diversity, with variations in diversity among different plant formations. The soil total potassium and nitrogen contents significantly affect the species diversity of the semi-natural plant communities, whereas the soil pH, soil total nitrogen content, organic matter content, available potassium content, and elevation significantly affect the community distribution. This study can provide important reference information for the construction of semi-natural or artificial plant communities on islands.

**Abstract:**

(1) Background: Islands are relatively independent and fragile ecosystems with unique habitats. Studying the relatively stable semi-natural plant communities on islands can enrich island vegetation data and provide an understanding of the factors affecting species diversity and distribution. (2) Methods: The semi-natural plant communities on Langqi Island (LI) in Fujian Province were sampled, and the redundancy method was used to analyze the correlation between species diversity, community distribution, and environmental factors. (3) Results: (i) There were 432 species of vascular plants, with 306 genera and 110 families; (ii) the semi-natural plant communities consisted of three vegetation types, 10 formations, and 10 clusters; (iii) the species diversity indices of the semi-natural plant communities presented a trend of gradually decreasing along the shrub, tree, and herb layers. Different formations varied in terms of the species diversity index; and (iv) the soil total potassium and nitrogen contents significantly affected the species diversity of the LI semi-natural plant communities, whereas the pH, soil TN content, organic matter content, AK content, and altitude significantly affected the community distribution. (4) Conclusions: Selecting appropriate tree species for mixed forests based on plant characteristics and regional conditions, together with regulating the soil nutrient content, can improve the species diversity of semi-natural plant communities.

## 1. Introduction

The characteristics and species diversity of plant communities have always been the focus of ecological research. Species diversity refers to the degree of diversity in the organization of species within a biological community. Indices such as the Margalef index, Simpson index, Shannon–Wiener index, and Pielou index are commonly used in community species diversity studies [1]. The species diversity of a plant community reflects its heterogeneity in terms of structural type, function, stability, and developmental stage and the relationship between different natural environments and plant communities [2]. Increasing species diversity enhances the functionality and stability of ecosystems [3]. Systems with higher species diversity are considered more efficient, productive, and sustainable [4]. Therefore, recording species diversity data of plant communities is crucial for the conservation and sustainable development of plant communities and ecosystems.

The maintenance of community species diversity has internal and external factors. Internal factors arise from differences in species biology and ecological characteristics, while external factors result from small-scale differences in community habitats, known as habitat heterogeneity [5]. Internal factors enable the coexistence of multiple species in the same habitat through the differentiation and complementarity of habitat resource utilization. External factors provide the conditions for the coexistence of multiple species. Habitat differentiation for specific plant communities mainly manifests in aspects such as light and soil conditions owing to relatively consistent large-scale climate conditions [5]. The current research on species diversity and its maintenance mechanisms is primarily focused on inland areas [6,7,8]. However, islands may influence plant diversity differently from that of the mainland owing to factors such as habitat area, habitat heterogeneity, habitat quality, and dispersal limitations [9,10]. Nevertheless, the mechanisms underlying the formation and maintenance of plant diversity in island systems remain unclear. Therefore, the following questions arise: (1) What are the unique characteristics of plant communities and species diversity on islands? (2) How are the mechanisms maintaining plant community species diversity on islands, and what are the environmental factors influencing species diversity and community distribution? The soil factors directly or indirectly affect vegetation growth and plant community species diversity [5,10,11], but their impact on plant diversity varies across different regions [10]. Therefore, further research is required to determine plant community species diversity on islands related to soil conditions and how the soil factors affect it.

Semi-natural communities refer to natural communities with mild human disturbances or communities formed through the natural growth, renewal, and succession of artificially planted vegetation [12,13]. Langqi Island (LI) is the fourth-largest island in Fujian Province; it is a coastal island that is rich in vegetation resources. The LI semi-natural plant communities (LISPCs) on the island formed through aircraft seeding or artificial afforestation during the last century, and the process was controlled by local policies, with relatively little human influence involved. The community understory includes rich and diverse plant life, with dominant species suited for survival in the community gradually emerging via community succession, renewal, and evolutionary processes, resulting in a relatively stable community structure [14].

We aimed to address the following primary questions: (1) What are the characteristics and species diversity of plant communities on islands? (2) What are the factors influencing plant community species diversity and species distribution on islands? To solve questions (1) and (2), the characteristics and species diversity of the LISPCs were analyzed and the environmental factors affecting their species diversity and distribution examined. This research can provide important reference information for the construction of semi-natural or artificial plant communities on islands.

## 2. Materials and Methods

### 2.1. Overview of the Study Area

Langqi Island is located on the north-central coast of Fujian Province (26°03′–26°07′ N, 119°32′–119°39′ E) and is under the jurisdiction of the Mawei District of Fuzhou City. The island is 15.3 km long from east to west and 8.1 km wide from north to south, with a coastline totaling 30 km and a total land area of 92 km^2^, of which land and tidal flats occupy 55 and 37 km^2^ and plains and mountains account for 66.36 and 33.64% of the land area, respectively. The island has a subtropical oceanic monsoon climate, with an average annual temperature of 19 °C, an average annual precipitation of 1600 mm, and a frost-free period of 326 days. The weather is warm and humid throughout all four seasons, and precipitation is abundant and concentrated. Northeasterly winds predominate, with the strongest winds reaching level 10 (i.e., storms). The island is mainly composed of aeolian sandy soil with poor water retention performance and low nutrient contents. The dominant vegetation type is subtropical evergreen broad-leaved forests with widely distributed species, including *Acacia confusa*, *Eucalyptus robusta*, *Celtis sinensis*, *Sageretia thea*, *Arthraxon hispidus*, *Bidens pilosa*, and *Digitaria sanguinalis.*

### 2.2. Setup of the Sample Plots and Field Surveys

The stratified random sampling method was selected, with 11 large mountains and a large *Celtis sinensis* forest on the island producing 12 sample plots in total (Figure 1). Previous research has shown that a sampling area of ≥1200 m^2^ is sufficiently representative for subtropical evergreen broad-leaved forests [15]. A preliminary survey was followed by stratification based on high-definition satellite maps and the existing geographical information, ensuring a sufficiently representative sample size [16]. Using the quadrat as a sampling unit and fixed-pattern distribution [17], 33 tree quadrats measuring 20 × 20 m were established, with four 5 × 5 m and four 1 × 1 m shrub quadrats at the four corners. A total of 132 shrub and 132 herb quadrats were examined in total.

Field surveys were conducted in the summer of 2022. Woody plants were classified into tree and shrub layers based on a height of 3 m. Plants with a diameter of less than 2.54 cm (1 inch) at breast height were considered shrubs and woody plants (seedlings) with heights not exceeding 30.5 cm (12 inches) herbs. The name, plant height, crown width, and plant quantity of the species in the tree and shrub layers were recorded for each quadrat, with the species name, plant height, coverage, and plant quantity for the herb layer and the diameter at breast height additionally recorded for the shrub layer. Specimens and photographs were obtained for plants that could not be identified on-site, and reference was then made to the Flora Fujianica [18], Flora of China [19], and other materials for identification. Invasive plants were determined from the Alien Invasive Flora of China by Ma Jinshuang [20].

A GPS was used to record the longitude, latitude, and altitude of each survey point. A gyrocompass was used to record the slope, aspect, and position of each slope (Appendix A) and the slope aspect recorded using a hierarchical system. The true north was set at 0° and eight aspect grades divided based on clockwise rotation over 45°, with each grade represented by a number. Higher numbers indicated more sun exposure [21]. The slope was similarly represented by Arabic numerals, with 1, 2, and 3 representing uphill, mid-slope, and downhill locations, respectively.

A five-point sampling method was used to sample the soil in each quadrat, with a soil auger of 5 cm in diameter used to acquire soil samples from depths of up to 25 cm. Samples were mixed and placed inside numbered self-sealing bags before transfer to the laboratory for processing. Samples were ground, sieved (0.15 mm), and stored for the subsequent experimental analysis.

### 2.3. Determination of the Soil Physicochemical Properties

Nine indicators were selected as the soil factors: organic matter; pH value; total (T) P, K, and N (TP, TK, and TN encompass the sum of various forms of phosphorus, potassium, and nitrogen in the soil, respectively); available (A) P, K, and N (AP, AK, and AN represent the phosphorus, potassium and nitrogen in the soil that can be absorbed by plants, respectively); and water-soluble salts [22,23]. All indicators were measured in triplicate. Organic matter was measured with an elemental analyzer, pH was measured with a pH meter, TP and AP were measured using the Mo-Sb colorimetric method, TK and AK were measured using a flame photometer, and TN was measured using the Kjeldahl determination method. AN was measured using titration, and water-soluble salts were measured using the gravimetric method.

### 2.4. Data Processing and Analysis

The families and genera of the seed plants in the study area were defined based on the Eight-order System of Angiosperms by Wu [24,25,26]. Ferns were categorized according to the classification system proposed by Qin [27]. The group-average clustering method was used to classify the plant community type. This method is commonly used in the classification of plant communities and is based on the species importance value (IV) of a community (rare species with IVs < 5 were eliminated) [28,29]. The equations used to calculate the IV and diversity index are provided in Table 1 [30,31].

One-way ANOVA was used with the least significant difference method in SPSS to calculate and analyze each community layer and the variations in the species diversity among different communities. The high spatial resolution GF-2 remote sensing data were imported into ArcGIS10.7, and the closest distance between the center coordinates of each quadrat and the sea was calculated. Bivariate correlation and collinearity analyses were performed using 14 environmental factors (of which 9 were soil-related and 5 were terrain-related), with the resulting VIF values for the various driving factors of <5 indicating that no collinearity relationship existed. Redundancy analysis (RDA) of the environmental factors and community species diversity and canonical correlation analysis (CCA) of the environmental factors and community distribution were performed using Canoco5.

## 3. Results

### 3.1. Species Composition

The LISPCs comprised 432 vascular plant species (including infraspecific units) from 306 genera and 110 families. Eight large families, including *Gramineae*, *Asteraceae*, and *Fabaceae*, dominated the vegetation on LI, all of which included multiple species. *Ficus* was the only genus observed with 10 or more species, and minor or very minor genera dominated. The 234 very minor genera showed significant advantages and accounted for 76.47% of the total, with some accounting for only one species, such as *Trachelospermum* and *Elaeocarpus*. 

### 3.2. Classification of Vegetation Type

Based on the overall plant IV situation of the 33 LISPCs (Appendix B), *Acacia confusa, Eucalyptus robusta*, and *Celtis sinensis* dominated the tree layer; *Plumbago zeylanica*, *Lantana camara*, and *Rhodomyrtus tomentosa* dominated the shrub layer; and *Arthraxon hispidus*, *Digitaria sanguinalis*, and *Bidens pilosa* were predominant in the herb layer. A quadrat-species IV matrix was then established, and cluster analysis was performed (Figure 2).

The Euclidean distance was set to 7.5 based on the clustering results and actual conditions, and the LISPCs were divided into 10 clusters (Table 2) with communities comprising three vegetation types: evergreen coniferous forests, evergreen broad-leaved forests, and deciduous broad-leaved forests. Evergreen coniferous forests are ecosystems dominated by needle-leaved, cone-bearing trees that retain their leaves (needles) throughout the entire year. Evergreen broad-leaved forests are characterized by trees with broad leaves that remain green throughout the year. Deciduous broad-leaved forests consist of trees with broad leaves that shed during a particular season, often in autumn. The evergreen coniferous forest included *Pinus massoniana* and *Cunninghamia lanceolata* formations, while the evergreen broad-leaved forest included seven formations: *Casuarina equisetifolia*, *Acacia confusa*, *Eucalyptus robusta*, *Casuarina equisetifolia* + *Acacia confusa*, *Acacia confusa* + *Heptapleurum ellipticum*, *Eucalyptus robusta* + *Acacia confusa*, and *Eucalyptus robusta* + *Casuarina equisetifolia*. The deciduous broad-leaved forest was composed of only one species: *Celtis sinensis*.

### 3.3. Analysis of Species Diversity

The Margalef index is related to the species quantity and total plant count in the sample community. The Shannon–Wiener index is used to explain the richness of species in the sample community, while the Simpson index reflects changes in the species quantity within the community. The Pielou evenness index indicates the evenness of species distribution within the community [1].

A comparison of the average diversity index values obtained for the various LISPC layers and overall community can be seen in Figure 3. The shrub layer exhibits the highest values for all four diversity indices (*p* < 0.01), indicating that this layer has the highest species diversity. The Margalef index (*R*) in the herb layer is the lowest among all the layers (*p* < 0.01), suggesting that the herb layer has the fewest species compared to the other layers. Both the Simpson index (*D*) and Shannon–Wiener index (*H*) in the tree and herb layers are lower than the overall values (*p* < 0.05 and *p* < 0.01), indicating lower complexity in these layers. The Pielou index (*J*) in the tree layer is the lowest among all the layers (*p* < 0.05), indicating the least even distribution of plant species in the tree layer.

The comparison of the species diversity indices for the various formations is illustrated in Figure 4. The results of the Margalef index indicate no significant differences between the *E. robusta* + *A. confusa*, *C. lanceolata*, and *A. confusa* + *H. ellipticum* formations. These three formations are significantly higher than the *C. equisetifolia* + *A. confusa* and *E. robusta* + *C. equisetifolia* formations (*p* < 0.05). This indicates that these three formations have the highest species richness, while the *C. equisetifolia* + *A. confusa* and *E. robusta* + *C. equisetifolia* formations have the lowest species richness, especially when *C. equisetifolia* is mixed with other trees. The Simpson indices for the *E. robusta* + *A. confusa* and *A. confusa* + *H. ellipticum* formations were significantly higher than those obtained for the *C. equisetifolia* and *A. confusa* formations (*p* < 0.05). This suggests that *A. confusa* mixed with *E. robusta* and *H. ellipticum* can enhance the species diversity of the community compared to the pure *A. confusa* forest. The Shannon–Wiener index for *E. robusta* + *C. equisetifolia* was significantly lower than those obtained for *A. confusa* + *H. ellipticum* and *E. robusta* + *A. confusa* (*p* < 0.05). This reveals that the species diversity in the *E. robusta* + *C. equisetifolia* formation is the lowest. The Pielou index for *A. confusa* + *H. ellipticum* was significantly higher than that of the *p. massoniana*, *C. equisetifolia*, and *E. robusta* formations (*p* < 0.05). This indicates that the distribution of species in the *A. confusa* + *H. ellipticum* formation is the most even, suggesting that mixed planting in the forest enhances species evenness compared to pure forests.

### 3.4. Analysis of the Environmental Factors

#### 3.4.1. Correlation between the Species Diversity and Environmental Factors

The results of the RDA analysis are shown in Figure 5. Two of the environmental factors’ (soil TK and TN) contents significantly affected the species diversity (*p* < 0.05). The soil TK positively correlated with the species diversity index for the shrub layer. The soil TN positively correlated with both the species diversity index for the entire community and the tree and herb layers but negatively correlated with that of the shrub layer. The two environmental factors jointly explained 19.8% of the species diversity, meaning that 80.2% of the factors affecting the species diversity in the LISPCs were still unknown.

#### 3.4.2. Correlation between the Formation Distribution and Environmental Factors

The CCA results indicated that five environmental factors significantly affected the formation distribution (*p* < 0.05): the pH, soil TN content, organic matter content, AK content, and altitude, which jointly explained 41.2% of the formation distribution (Figure 6). The first axis mainly reflects the gradient changes in the distribution of LISPCs, with variations in altitude, organic matter content, TN content, and pH observed. The organic matter content, TN content, and pH increased as the altitude decreased. The second axis is assumed to mainly reflect the gradient change in the AK content, which decreases from bottom to top in the figure. 

The classification results for the vegetation type show that the *Pinus massoniana* formations were distributed at high altitudes where soils are acidic and the TN, organic matter, and AK content are low. The *C. lanceolata* formations were distributed at lower altitudes in areas with higher soil AK contents. The *C. equisetifolia*, *A. confusa*, *A. confusa* + *H. ellipticum*, *E. robusta*, and *E. robusta* + *A. confusa* formations were mostly distributed at higher altitudes in acidic soil with lower TN and organic matter contents but a higher AK content. The *C. equisetifolia* + *A. confusa*, *E. robusta* + *C. equisetifolia*, and *C. sinensis* formations were distributed at lower altitudes in less acidic soils, higher TN and organic matter contents, and lower AK content. The CCA ranking and group-average clustering results are consistent and better reflect the variations in the environmental factors that affect the various semi-natural plant formations.

## 4. Discussion

### 4.1. Characteristics

The species compositions of the plant communities on the island are affected by several factors, such as environmental change, human interference, interspecific competition, and geographical isolation [9]. The LISPCs are dominated by herbaceous plants and shrubs, with annual herbs accounting for a significant proportion of the total. This is attributed to the annual plants, with their extremely strong vitality and reproductive abilities, growing and rapidly developing under the strong winds, high temperatures, and an arid environment. For instance, the dominant LI families include Gramineae (35 species) and Asteraceae (29 species), both of which have adapted to the island habitat, with Asteraceae achenes suited to wind dispersal and Gramineae leaves showing heat and drought resistance [28]. In the LISPCs, the dominant layer is the shrub layer, with the highest values for various diversity indices. The key species in this layer include *M. repandus*, *S. thea*, *S. china*, and *E. oldhamii*, among others. These strongly adaptive species often thrive in unique island habitats and become dominant [32].

The predominant vegetation type in the LISPCs is evergreen broad-leaved forest, which comprises zonal vegetation in the subtropical region. The community structure is primarily tree–shrub–grass, although tree–grass was also observed. The dominant species in the tree layer included *A. confusa*, *E. robusta*, *C. sinensis*, and *C. equisetifolia*. *A. confusa* and *C. equisetifolia* are also the common dominant species on several neighboring islands [33,34,35]. Before 1949, forceful sandstorms created extensive wind-formed sand dunes between LI and Dongshan Island. After the 1960s, aircraft seeding of *A. confusa* and *C. equisetifolia* was performed as a means of controlling sand erosion, rendering the IVs of these species high. However, *C. sinensis* was the original dominant species unique to the LI tree layer. This species thrives well in the barren, loose, and highly permeable coastal aeolian sandy soil, reflecting its potential as a windbreak. Planting *C. sinensis* with *C. equisetifolia* enhances the stability of *C. equisetifolia* and alleviates the degradation of windbreaks formed with this species [36]. 

### 4.2. Species Diversity and Environmental Factors

The results of the RDA analysis show that the soil TN and TK significantly affected the species diversity index of the LISPCs. Nitrogen is an essential nutrient required for the growth and development of plants [37]. Coastal areas are often subjected to high temperatures and precipitation, resulting in the loss of large amounts of N from island soils [38]. Island plant communities are often N-limited during growth, since plants can only absorb N in soluble mineral form [39]. Plants that have access to sufficient N absorb more N, leading to improved P, K, and Ca absorption. The more nutrients that the plants can obtain in each layer, the better their growth conditions and the more complex the community [29]. Therefore, increasing the TN can help improve plant diversity to a certain extent. In this study, the species diversity indices of the overall community and tree and herb layers was positively correlated with the TN, supporting the existing view. However, excessive soil N content leads to rapid increases in the biomass of the dominant species and intensifies interspecific competition, inhibiting the growth of other plants and leading the community structure toward homogenization [40]. This has a particularly significant impact on the shrub layer [41]. 

The results of the CCA analysis show that the soil TN content, pH, and organic matter content significantly affected the plant distribution of the LISPCs. A significant correlation that was observed between the soil TN content and the distribution of *A. confusa* formations might be related to the N-fixing ability of *A. confusa*, a leguminous plant. The soil TN content is dependent on the relative intensity of the soil organic matter accumulation and decomposition. In turn, the soil N affects the input of soil organic matter by affecting crop growth, leading to differences in the soil organic matter content [42]. In addition, the loss of N accelerates the loss of some other elements, resulting in soil acidification. The soil pH affects the availability of nutrients because of mineralization [43]. The *P. massoniana*, *C. equisetifolia*, among other formations, were distributed on soils with strong acidity, while *C. lanceolata* and *C. equisetifolia* + *A. confusa* (among other formations) were distributed on soils with weak acidity.

In this study, the TK was positively and negatively correlated with the altitude and slope position, respectively. The total phosphorous content is an indicator of the potential supply of soil K and is also a reflection of the degree of soil weathering. Aeolian sandy soils in high-altitudinal areas with strong winds tend to have higher TK contents [44]. Potassium also has the effect of endowing plants with cold resistance; the higher the altitude, the larger the K content in plant branches and leaves. Organic K is released from litter in inorganic form, thereby increasing the soil TK content [45]. In addition, K has strong mobility in the soil and is easily leached and lost lower on the slope, resulting in a reduced soil TK content [46]. The species diversity index of the shrub layer is greatly affected by the TK content. Shrub growth is boosted by a high soil K content [47], which is consistent with the results of this study. 

The CCA results indicated that the soil AK and altitude are also significant factors affecting the distribution of LISPCs, with changes in the altitude gradient affecting the concurrent variations in habitat conditions such as humidity, light, and soil, which, in turn, directly affect the distribution patterns of the plant formations [48]. The soil AK content significantly correlated with the distribution of *B. pilosa* and some Gramineae plants. *B. pilosa* is distributed in areas with a higher soil AK content, possibly because Asteraceae plants usually have strong allelopathic effects. A higher soil AK content facilitates plant synthesis with large amounts of defensive allelopathic substances such as phenols [49]. The soil AK is easily transmitted and undergoes vertical diffusion. The roots of some Gramineae plants (such as *A. hispidus* and *D. sanguinalis*) penetrate only the shallow soil layer. The strong mobility of AK helps alleviate the competition for K among herbaceous plants in the same soil layer, which is beneficial to their growth [50]. 

The *A. confusa* + *H. ellipticum* and *E. robusta* + *A. confusa* formations showed higher overall diversity indices. Higher soil TN and TK contents were observed for the *A. confusa* + *H. ellipticum* and *E. robusta* + *A. confusa* formations. However, the mixed planting of *A. confusa* with *E. robusta* and *C. equisetifolia* enhanced the community species diversity compared to pure forests. Tree layers that are composed of mixed forests contain many types of plants and large ecological niche variations, leading to full resource utilization [51]. In contrast, the species composition of pure forests is simple, with the singular stress caused by the tree layer reducing the number of understory plants. Moreover, the existence of fewer plants reduces the provision of sufficient litter for microbial decomposition. Consequently, worse habitat conditions are observed in pure forests compared to mixed forests, further reducing the species diversity indices of pure forests [29].

A study of the diversity of forest communities on Dalian Island showed that the overall *H* and *D* of the *A. confusa* + *C. equisetifolia* forest formations were significantly higher than those of the *A. confusa* and *C. equisetifolia* forests. Thus, they suggested that the biological species of the community and the sizes of the Simpson and Shannon–Wiener indices are larger for mixed forests compared to pure forests [28]. Unlike Dalian Island, mixed forests that contain *C. equisetifolia,* such as the *E. robusta* + *C. equisetifolia*, *C. equisetifolia* + *A. confusa*, and *C. equisetifolia* formations*,* have low overall diversity indices, especially in the shrub and herb layers. The reason was the action of *C. equisetifolia,* which inhibits the growth of other plants in the forest by releasing allelopathic substances, causing the understory shrubs and herbs to become rare and resulting in reduced diversity [52,53]. The diversity indices of the *P. massoniana* formations are also lower, especially various species diversity indices in the herb layer, which are at relatively low levels. The reason might be that the herb layer of the *P. massoniana* forest is dominated by *D. pedata*. *D. pedata* forms a dense layer under the forest, where it intercepts large amounts of litter from the canopy trees, affecting the litter decomposition and nutrient recycling processes of the forest. In addition, *D. pedata* is allelopathic in nature [54], inhibiting the growth of other species in the herb layer and greatly reducing the species diversity. 

The stand canopy density and presence of invasive plants might also have a certain impact on diversity. A higher stand canopy density reduces the light within the stand, changing the utilization potential of light by the plants, thereby reducing the species diversity in the shrub and herb layers [55]. A community with a single dominant species forms when the number of invasive plants reaches a certain threshold, which also decreases the plant diversity [56]. Hence, further studies are needed to investigate the related effects of canopy density and invasive plants on plant diversity.

## 5. Conclusions and Recommendations

A study related to the characteristics and species diversity of semi-natural plant communities on Langqi Island revealed that: The LISPC distribution was affected by the coupling of various factors rather than a single factor. The heterogeneity of the island habitat and the biological characteristics of the plants themselves were the main reasons affecting their distribution. The main reasons for the low overall diversity index of these formations in LISPCs is the soil TN and TK contents, while the allelopathic nature of some plants in the formations also affect diversity. The presence of mixed forest may increase the number of biological species and improve species diversity of the plant communities to a certain extent, with artificial thinning promoting the conversion of single-species to mixed forest. In the LISPCs, the formations with mixed planting of *A. confusa, H. heptaphyllum*, and *E. robusta* exhibited higher species diversity. The mixed planting of *C. sinensis* and *C. equisetifolia* enhanced the stability of *C. equisetifolia*, alleviating the degradation of a windbreak. The relationship between species quantity and species diversity is not simply linear. The level of species diversity cannot be improved by merely increasing the types and quantities of plants in the tree layer. This is because the requisite environmental conditions for growth: altitude, soil, and light vary in different plant communities. Therefore, it is necessary to adopt the principle of planting suitable trees in the right location based on their characteristics and regional status while promoting the long-term stable development of communities by supplementary planning.

## Figures and Tables

**Figure 1 biology-13-00011-f001:**
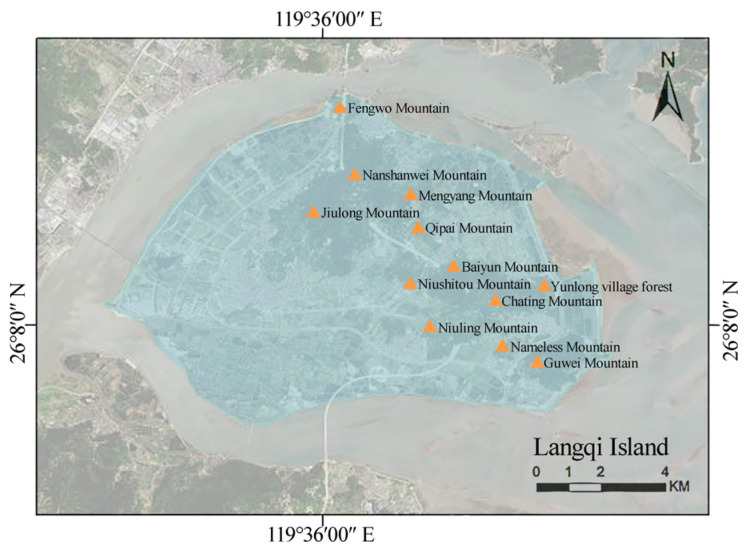
Location of Langqi Island (LI), and distribution of the sample plots [15,16].

**Figure 2 biology-13-00011-f002:**
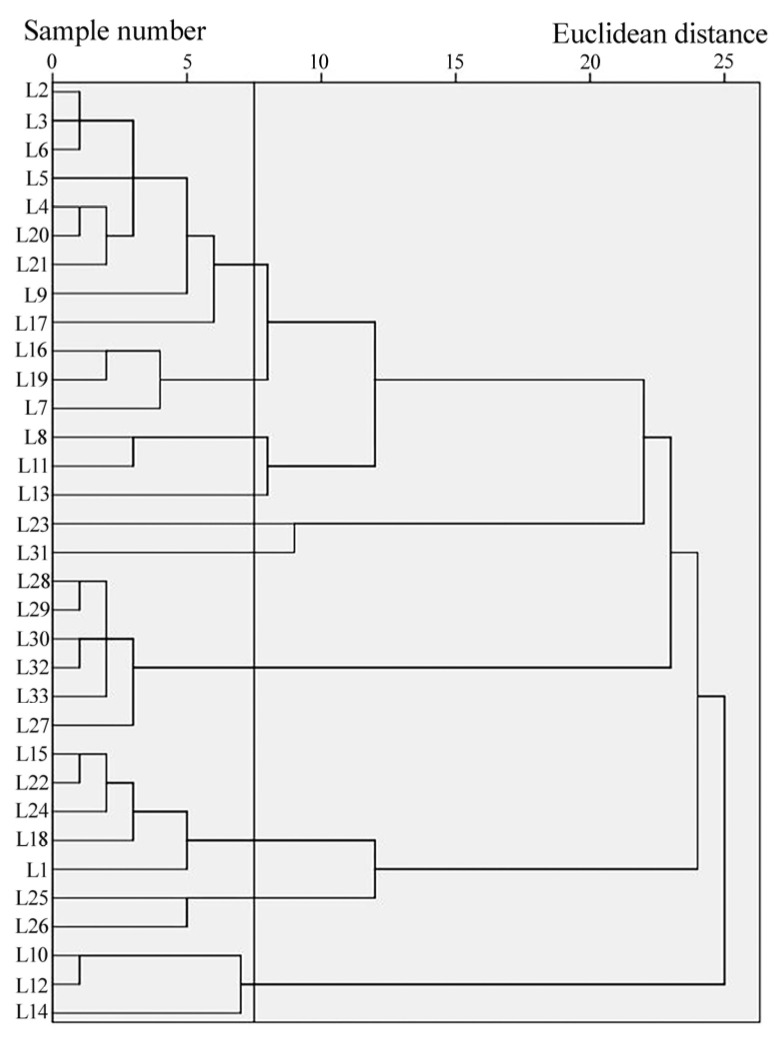
Average clustering diagram for Langqi Island semi-natural plant community (LISPC) formations.

**Figure 3 biology-13-00011-f003:**
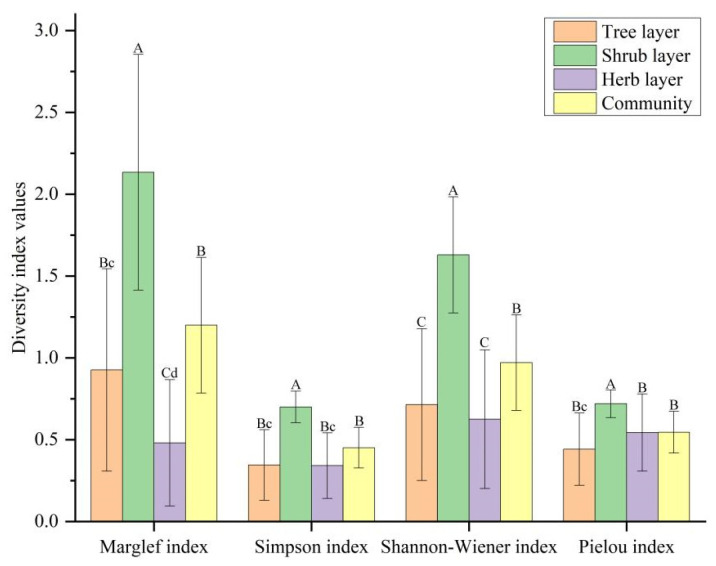
Comparison of the average diversity index values for the various LISPC layers and the overall community (mean ± standard deviation). The letters themselves have no meaning; they represent the results of group comparisons. If two groups do not have the same letter, there is a significant difference between them. Lowercase letters indicate significant differences (*p* < 0.05), and uppercase letters indicate extremely significant differences (*p* < 0.01).

**Figure 4 biology-13-00011-f004:**
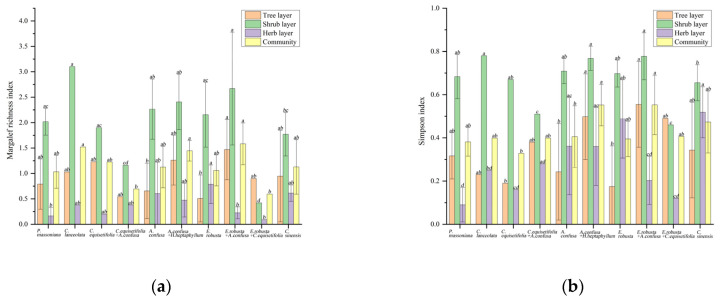
Species diversity indices for the various formations: the Margalef index (**a**), Simpson index (**b**), Shannon–Wiener index (**c**), and Pielou index (**d**). Lowercase letters indicate significant differences at *p* < 0.05.

**Figure 5 biology-13-00011-f005:**
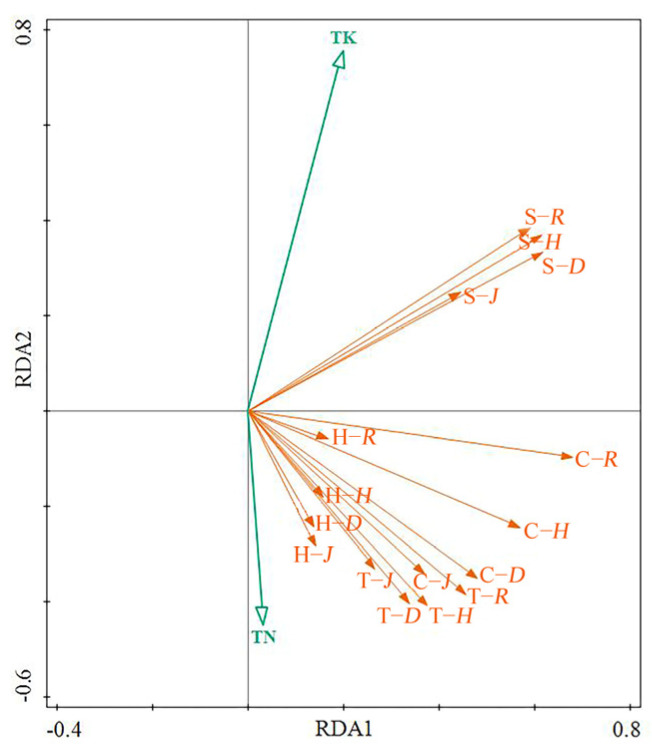
Redundancy analysis (RDA) of the species diversity and environmental factors. TK: soil TK content; TN: soil TN content; T−*R*: Margalef index for the tree layer; T−*D*: Simpson index for the tree layer; T−*H*: Shannon–Wiener index for the tree layer; T−*J*: Pielou index for the tree layer; S−*R*: Margalef index for the shrub layer; S−*D*: Simpson index for the shrub layer; S−*H*: Shannon–Wiener index for the shrub layer; S−*J*: Pielou index for the shrub layer; H−*R*: Margalef index for the herb layer; H−*D*: Simpson index for the herb layer; H−*H*: Shannon–Wiener index for the herb layer; H−*J*: Pielou index for the herb layer; C−*R*: community Margalef index; C−*D*: community Simpson index; C−*H*: community Shannon–Wiener index; C−*J*: community Pielou index.

**Figure 6 biology-13-00011-f006:**
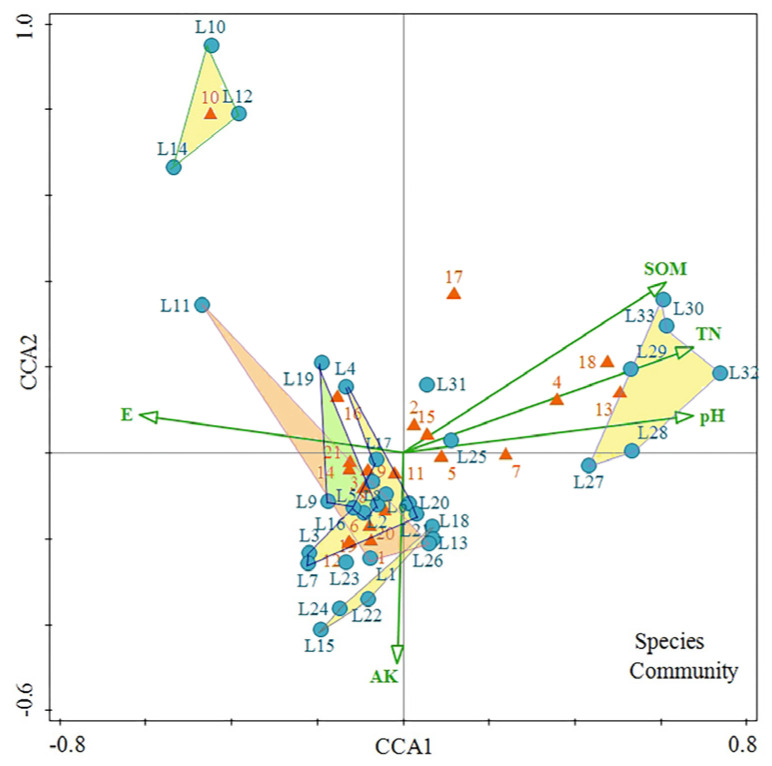
Typical correlation analysis for the community quadrats and environmental factors. TN: soil TN content; pH: soil pH; SOM: soil organic matter content; E: altitude; AK: soil AK content; 1: *Eucalyptus robusta*; 2: *Smilax china*; 3. *Heptapleurum heptaphyllum*; 4. *Elaeagnus oldhamii*; 5. *Bidens pilosa*; 6. *Arthraxon hispidus*; 7. *Melia azedarach*; 8. *Zanthoxylum nitidum*; 9. *Digitaria sanguinalis*; 10. *Pinus massoniana*; 11. *Casuarina equisetifolia*; 12. *Rubus hirsutus*; 13. *Celtis sinensis*; 14. *Oplismenus undulatifolius*; 15. *Sageretia thea*; 16. *Cyperus rotundus*; 17. *Dianella ensifolia*; 18. *Liriope spicata*; 19. *Rubus corchorifolius*; 20. *Mallotus repandus*; 21. *Acacia confusa*.

**Table 1 biology-13-00011-t001:** Calculation equations for the species importance value (IV) and species diversity index.

Species IV and Diversity Index	Calculation Equation	Explanation
Tree layer IV	IV_1_ = (*Ar* + *Pr* + *Fr*)/3	*Ar* ‘relative abundance’ = (number of plants of a particular species)/(total number of plants of all species) × 100% *Pr* ‘relative dominance’ = (basal area of a particular species)/(total basal area of all species) × 100% *Fr* ‘relative frequency’ = (frequency of a particular species)/(total frequency of all species) × 100% *Cr* ‘relative coverage’ = (sum of coverage of all individuals of a particular species)/(total coverage of all species) × 100%
IV of the shrub and herb layers	IV_2_ = (*Ar* + *Cr* + *Fr*)/3	
Margalef index	*R =* (*S* − 1)/ln*N*	*S* is the number of species in the quadrat and *N* is the sum of the numbers of all species. *N_i_* is the number of individuals of the *i*th species, *i* = 1, 2, 3, …S, and *P_i_* is the species IV
Simpson’s dominance index	$D=1-\sum_{i=1}^{S}P_{i}^{2}$	
Shannon–Wiener diversity index	$H=-\sum_{i=1}^{S}P_{i}lnP_{i}$	
Pielou uniformity index	*J* = *H* /ln*S*	
Overall community diversity	*W* = 0.5*W_T_* + 0.3*W_S_* + 0.2*W_H_*	*W* is the overall diversity index of the community; *W_T_*, *W_S_*, and *W_H_* are the diversity indices of the tree, shrub, and herb layers, respectively

**Table 2 biology-13-00011-t002:** LISPC vegetation types.

Vegetation Type	Formation	Cluster
Evergreen coniferous forest	*Pinus massoniana*	*Pinus massoniana* − *Rhodomyrtus tomentosa* + *Rosa laevigata* + *Melastoma malabathricum* − *Dicranopteris pedata* + *Cyperus rotundus*
	*Cunninghamia lanceolata*	*Cunninghamia lanceolata* − *Heptapleurum delavayi + Begonia cucullata* − *Pteris semipinnata* + *Arthraxon hispidus* + *Dicranopteris pedata*
	*Casuarina equisetifolia*	*Casuarina equisetifolia* − *Rubus corchorifolius* − *Pteris semipinnata* + *Arthraxon hispidus*
	*Casuarina equisetifolia* + *Acacia confusa*	*Casuarina equisetifolia* + *Acacia confusa* − *Sageretia thea* − *Bidens pilosa*
	*Acacia confusa*	*Acacia confusa* − *Elaeagnus oldhamii* + *Eleutherococcus trifoliatus* + *Rubus hirsutus* − *Arthraxon hispidus* + *Oplismenus undulatifolius*
Evergreen broad-leaved forest	*Acacia confusa* + *Heptapleurum ellipticum*	*Acacia confusa* + *Heptapleurum heptaphyllum* − *Sageretia thea* + *Murraya exotica* + *Mallotus repandus* − *Digitaria sanguinalis* + *Arthraxon hispidus*
	*Eucalyptus robusta*	*Eucalyptus robusta* − *Rubus hirsutus* + *Miscanthus floridulus* + *Mallotus repandus* − *Bidens pilosa* + *Arthraxon hispidus*
	*Eucalyptus robusta* + *Acacia confusa*	*Eucalyptus robusta + Acacia confusa* − *Rhodomyrtus tomentosa + Smilax china* − *Digitaria sanguinalis* + *Oplismenus undulatifolius* + *Chrysanthemum indicum*
	*Eucalyptus robusta* + *Casuarina equisetifolia*	*Eucalyptus robusta* + *Casuarina equisetifolia* − *Bidens pilosa* + *Cyperus rotundus*
Deciduous broad-leaved forest	*Celtis sinensis*	*Celtis sinensis* − *Plumbago zeylanica + Eleutherococcus trifoliatus + Elaeagnus oldhamii*−*Liriope spicata + Equisetum ramosissimum*

## Data Availability

The data presented in this study are available on request from the corresponding author. The data are not publicly available due to privacy restrictions.

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
