# Peer review of "Characteristics and Species Diversity of Semi-Natural Plant Communities on Langqi Island"

_biology, 2023, doi:10.3390/biology13010011_

Round 1

Reviewer 1 Report

Comments and Suggestions for Authors

1. Please some information in manuscript, the suggestions in the text.

2. Some corrections are provided in text.

3. please add some references in figure and in text that is comments in text.

Reviewer 2 Report

Comments and Suggestions for Authors

The manuscript studied the relatively stable close-to-nature plant communities on island to provides understanding of the factors affecting species diversity and distribution. By using the 11 plots data to analyzed the species composition, vegetation type, and species diversity of the close-to-nature plant communities and environmental factorson Langqi Island in Fujian province. The manuscript is valuable to protect the island plant diversity and function of local ecosyetem.

Below, I provide the suggestions for improvement.

1.      The title is not suitable to the main scientific questions of this manuscript, and the results and conclusion is not relative to close-to-nature plant community, what is the local nature plant community?

2.      Totally, this introduction part of manuscript is full of trivial details, must be concise, the scientific question of this manuscript is unclear.

3.      The analysis in the results is simple, and the figure part is indistinct.

4.      The Discussion part is also full of trivial details, same to the results,

5.      the conclusions part is short of the valuable suggestions.

6.      The references is short of the many important current researches of the mechanism of plant community stability and species diversity maintenance.

Major revision recommended.

Reviewer 3 Report

Comments and Suggestions for Authors

The article is interesting and gives a picture of the plant communities present on the island under study which tries to relate the environmental parameters to the different communities. However, some aspects need to be explored further. In particular, the concept of communities close to nature (CTN) should be better clarified. I could understand that these communities are those closest to the maximum evolution of the vegetation for that territory, something similar to the peak of the vegetation series of the European authors or the climax of the American authors. On the other hand, in recent literature there is more discussion of forestation close to nature or restoration close to nature with native species and I believe we fall into this area; but in the case under study there are also alien species such as Eucalyptus which somewhat contradict the initial assumption. Therefore the authors should clarify better what they mean by CTN in the introduction.

Furthermore, it would be appropriate in the results to make a more complete description of the plant communities identified, a sort of sheet that summarizes floristic composition, ecological needs and distribution in the territory which is useful for the purposes of a synthesis after the numerous statistical checks on communities and environmental parameters taken into account consideration.

Some other suggestions in the attached file.

Comments on the Quality of English Language

English is not my language but the manuscript seems good regarding English writing

Round 2

Reviewer 3 Report

Comments and Suggestions for Authors

In this new version the manuscript has been significantly improved and can therefore be published in the journal "Biology"

I only reiterate that although the names of the plants are reported in full in Table 2 and in Appendix B it would be more elegant and appropriate to include them in full the first time a taxon appears in the text. This was done by the authors but only partially.

Comments on the Quality of English Language

I am not of English language but it seems to me good.

Author Response

Thank you for your comments and we have added the full name when it appears first in the text. Please see the following changes in the first paragraph of page 6.

The evergreen coniferous forest included Pinus massoniana and Cunninghamia lanceolata formations, while the evergreen broad-leaved forest included seven formations: Casuarina equisetifolia, Acacia confusa, Eucalyptus robusta, Casuarina equisetifolia + Acacia confusa, Acacia confusa + Heptapleurum ellipticum,Eucalyptus robusta + Acacia confusa, and Eucalyptus robusta + Casuarina equisetifolia. The deciduous broad-leaved forest was composed of only one species: Celtis sinensis.